# Enhancing Sustainable Transportation: AI-Driven Bike Demand Forecasting in Smart Cities

**Malliga Subramanian [1], Jaehyuk Cho [2,\*], Sathishkumar Veerappampalayam Easwaramoorthy [2], Akash Murugesan [1] and Ramya Chinnasamy [1]**

1   Department of Computer Science and Engineering, Kongu Engineering College, Erode 638060, India; mallisenthil.cse@kongu.edu (M.S.); akashm006@outlook.com (A.M.); ramya.me@gmail.com (R.C.)
2   Department of Software Engineering, Jeonbuk National University, Jeonju 54896, Republic of Korea; sathish@jbnu.ac.kr
\*   Correspondence: chojh@jbnu.ac.kr

**Abstract:** Due to global ecological restrictions, cities, particularly urban transportation, must choose ecological solutions. Sustainable bike-sharing systems (BSS) have become an important element in the worldwide transportation infrastructure as an alternative to fossil-fuel-powered cars in metropolitan areas. Nevertheless, the placement of docks, which are the parking areas for bikes, depends on accessibility to bike paths, population density, difficulty in bike mobility, commuting cost, the spread of docks, and route imbalance. The purpose of this study is to compare the performance of various time series and machine learning algorithms for predicting bike demand using a two-year historical log from the Capital Bikeshare system in Washington, DC, USA. Specifically, the algorithms tested are LSTM, GRU, RF, ARIMA, and SARIMA, and their performance is then measured using the MSE, MAE, and RMSE metrics. The study found GRU performed the best, with RF also producing reasonably accurate predictions. ARIMA and SARIMA models produced less accurate predictions, likely due to their assumptions of linearity and stationarity in the data. In summary, this research offers significant insights into the efficacy of diverse algorithms in forecasting bike demand, thereby contributing to future research in the field.

**Keywords:** bike-sharing systems; docks; ARIMA; SARIMA; LSTM; GRU; performance metrics; demand prediction

## 1. Introduction

Over the past few decades, shared bikes have gained significant attention as a sustainable urban transportation option [1]. Bike-sharing systems (BSS) offer a green alternative for short-distance travel, reducing carbon emissions and improving last-mile connectivity to public transit [2]. Notably, during the COVID-19 pandemic, BSS emerged as a resilient mode, addressing concerns about overcrowding in public transit. The integration of BSS into intelligent urban traffic systems has several benefits [3]. For instance, it enhances the efficiency of public transportation, reducing road traffic congestion [4]. However, one significant challenge in BSS is the imbalanced distribution of bikes, affecting users' riding experience. Research has shown that urban centers have a higher bike concentration than rural areas, and the demand for bikes experiences periodic peaks during work commute times [5]. The demand patterns in BSS are influenced by both spatial and temporal dependencies. Spatial aspects involve the varying demand across different areas and the geographic distribution of bike stations. Temporal dependencies refer to fluctuations in demand throughout the day, week, or year. Understanding these dependencies is crucial for operators to identify high-demand areas and optimize resource allocation, such as redistributing bikes or expanding stations. To tackle the spatial and temporal disparities, methods for bike redistribution have been formulated. These approaches involve the utilization of trucks or trailers to reposition bicycles within the urban area [6,7]. To

enhance service effectiveness and minimize the expenses associated with redistribution, researchers have harnessed historical bike usage information to precisely forecast upcoming demand [8,9]. The accurate prediction of bike demand is fundamental to effective redistribution and overall BSS performance.

The development of BSSs insists on the significant role played by technological advancements. Also, the success of these systems is dependent on the solution to three issues: (a) the number of stations and where they are located; (b) the capacity of the stations; and (c) the distribution of bikes. Estimating the demand placed on the system is necessary in order to find a solution to these issues. An in-depth analysis of bike-sharing research published between 2010 and 2018 reveals that demand estimation is one of the most swiftly developing trends in the field of bike-sharing research worldwide [10]. Considering the significance of BSS in urban transportation, precise demand forecasting plays a vital role in ensuring efficient bike rebalancing during daily operations. Both traditional and machine learning methods have been the subject of numerous research efforts to develop frameworks that can accurately anticipate citywide bike demand. The precise prediction of bike demand enables operators to optimize bike allocation and resource management, leading to enhanced customer satisfaction and reduced operational expenses.

Time series modeling and regression analysis are two common approaches to bike demand forecasting. Time series models are used to model and forecast data that changes over time, such as hourly or daily bike demand. Time series models can capture patterns such as trends, seasonality, and cyclicality in the data and use them to make predictions. Regression analysis, on the other hand, is used to model the relationship between one or more predictor variables and a response variable. In the context of bike demand forecasting, regression analysis can be used to model the relationship between bike demand and various factors such as weather, time of day, day of the week, and holidays. Regression models can capture the effects of these factors on bike demand and use them to make predictions. Both time series modeling and regression analysis have their strengths and weaknesses, and the choice of approach depends on the specifics of the problem at hand. Time series models are useful when the focus is on forecasting future bike demand based on historical patterns, while regression analysis is useful when the focus is on understanding the factors that affect bike demand and how they can be leveraged to improve forecasting accuracy. In this work, we use both types of algorithms to predict the demand for bikes during the next day and next month.

The present research focuses on addressing the following questions:

1. How can historical bike usage data be effectively utilized to predict future demand in a BSS, and which time series forecasting and regression algorithms are most suitable for predicting bike demand in a BSS? Can we generalize the models for different BSSs?

The challenge involves utilizing historical bike usage data to accurately forecast the future bike demand within a BSS. This encompasses understanding the patterns, trends, and potential influencing factors that drive bike demand fluctuations over time and generalize the models.

2. How can the integration of temporal factors, such as day of the week, time of day, and seasonality, improve the accuracy of bike demand predictions using time series and regression algorithms?

With a focus on harnessing temporal dynamics, such as day of the week, time of day, and seasonal influences, this article aims to uncover novel insights that optimize the predictive accuracy of time series and regression algorithms.

To address these questions, we use the dataset that represents a two-year historical log from the Capital Bikeshare system, Washington, DC, USA [11]. To predict the demands for the bike, we propose to use regression models such as Random Forest and time series models such as ARIMA, SARIMA, LSTM, and GRU. In addition, we test the generalizability of the developed models using another dataset. This work not only seeks to advance the field of urban mobility and sustainable transportation but also promises practical

implications for optimizing bike allocation, ensuring user satisfaction, facilitating more efficient urban planning strategies, and so on.

The main contributions of the proposed work include:

1. Conduct an exploratory analysis of trends, patterns, outliers, and unsettled points in bike prediction
2. Analyze the fine-grained temporal factors, such as the day of the week, time of day, and seasonality, which play a crucial role in shaping bike demand patterns in urban environments and utilize AI techniques to capture and leverage these patterns for better forecasting.
3. Develop AI-driven forecasting models tailored for bike demand prediction using time series and regression algorithms and evaluate their performance using MAE, RMSE, and MSE.
4. Validate the developed models against a new dataset: the London Bike Sharing System.

The rest of the article is orchestrated as follows: Section 2 provides a comprehensive review of the existing works related to the BSS. In Section 3, a description of the dataset and an exploratory data analysis are presented. A brief review of the time series and regression models used in the present work is also given in Section 3. In Section 4, we detail the experimental settings we used and the results of the developed models. A discussion on the performance of the proposed models is presented in Section 5. Finally, we provide a conclusion and scope for further extension of our work in Section 6.

## 2. Literature Review

Various empirical studies have employed diverse predictive models to forecast bike-sharing demand. These models typically incorporate historical data and multiple external factors, such as weather conditions, temporal details, and spatial information. A classification system for predicting BSS [12] is introduced based on the specific data formats obtained from both docked and dockless BSS. In this section, we provide a comprehensive overview of different models utilized for predicting BSS.

In the study conducted by Sathishkumar et al. [13], a predictive model for bike-sharing demand was developed using a machine learning approach. Data from a BSS in Bangalore, India, was used, and the most significant predictors of bike demand were found to be temperature and humidity. The effectiveness of a machine learning approach for predicting bike-sharing demand was demonstrated, and the importance of considering external factors such as weather and holidays was highlighted. The findings can inform decision-making for bike-sharing operators, urban planners, and policymakers to develop more effective and sustainable transportation systems. In the study by Brownlee et al. [14], the authors have explored common data normalization techniques used in machine learning. The benefits and drawbacks of each technique were also discussed, along with guidance on when to use each based on the dataset characteristics. The study by Box et al. [15] aimed to develop a framework for time series analysis known as the Box–Jenkins methodology, and the methodology consists of three stages: model identification, parameter estimation, and model checking. The authors demonstrated the effectiveness of the approach through various case studies, including modeling the demand for a particular brand of beer and predicting the number of airline passengers. The study contributed significantly to the development of time series analysis and provided a useful resource for practitioners and researchers in the field.

In the study conducted by Hyndman et al. [16] a new approach to forecasting time series known as the "forecast" package was introduced. The package provides a range of functions for forecasting, including ARIMA models, exponential smoothing models, and seasonal decomposition methods. The authors also examined methods for evaluating and selecting models for use with various kinds of time series data. The effectiveness of the approach was demonstrated through a range of case studies, including forecasting tourism demand, electricity demand, and stock prices. The authors of [17,18] examined automatic forecasting methods for handling a considerable volume of univariate time series,

commonly required in business and other fields. Gao et al. [8] developed four distinct machine-learning models to predict consumer demand for bike sharing in Seoul. The researchers went beyond weather-related features and incorporated additional variables, such as air pollution, traffic information, COVID-19 cases, and socio-economic factors, to enhance the accuracy of their predictions. The dataset used in the study was obtained from the Seoul Public Data Park website, encompassing the counts of public bike rentals in Seoul, South Korea, throughout the year 2020. Among the 29 features categorized into six groups, the weather, pollution, and the COVID-19 outbreak features proved to be the most influential in the model's prediction performance.

Schuijbroek et al. [7] delved into BSSs that have gained widespread popularity, with installations in numerous cities worldwide. The study highlighted that the primary operational cost driver for these systems is the bike rebalancing process, aiming to maintain a suitable number of available bikes and open docks for users. To address this challenge, the researchers proposed a new heuristic approach, prioritizing clustering first and routing second. This method addresses a clustering challenge of polynomial size, taking into account both the feasibility of service levels and the approximate routing costs concurrently. Conversely, Philipp Probst et al. [19] constructed a model using the RF algorithm. This model encompasses numerous user-adjustable hyperparameters, including the count of randomly drawn observations per tree (with or without replacement), the number of randomly selected variables for each split, the criteria for splitting, the minimum necessary samples within a node, and the overall count of trees forming the forest. Zhou [20] examined Chicago's spatiotemporal riding pattern using huge BSS data from July to December 2013 and 2014 and used the fast greedy algorithm to find biking flow spatial communities using a bike flow similarity graph. This work found weekday and weekend travel patterns and customer and subscriber travel trends in the noisy large data. Using the hierarchical clustering method, the authors also looked at the temporal demands for bikes and docks.

In a study by [21], researchers introduced a stacking model to predict variations in public bicycle traffic flow using real-world data. XGBoost was employed to train the models and identify factors influencing public bicycle traffic flow. The features utilized in this study include time, space, history, and weather information. To further enhance the analysis, the authors utilized the K-Medoids algorithm to group bike stations. This was accomplished by creating a novel station correlation matrix that relies on the distance between the stations. Another study conducted by Zhao et al. [5] proposed a novel hyper-clustering approach to enhance a spatiotemporal deep neural network for traffic prediction in BSSs. This innovative approach captured mobility trends among individuals and clusters, resulting in improved accuracy in predicting the number of available bikes. The experimental findings showed that the spatiotemporal deep neural network model, with hyper-clustering, surpassed previous approaches in accurately predicting bike demand. Dastjerdi and Morency [22] researched short-term forecasting, specifically predicting shared bike demand in Montreal 15 min ahead, using a deep learning approach. The study started by identifying six communities within the bike-sharing network using the Louvain algorithm based on a set of bike trips. To forecast pickup demand in each community, LSTM-based architectures were utilized. As a benchmark, a univariate ARIMA model was also employed for comparison. In a work by Lee and Ku [23], an RNN model was proposed, incorporating a dual attention mechanism to extract both spatial and temporal features. The attention mechanism effectively determines and weights all location features in the time series data, facilitating the learning of mutual correlations. Additionally, a random walk mechanism was incorporated during the preprocessing stage to maintain local relationships between bike stations, making the model more adaptable to local location changes across different stations.

To summarize, the literature survey revealed that this is an actively researched area, with a range of models and techniques being used. Some studies have focused on using traditional regression models, while others have explored the effectiveness of more complex machine learning models and time series models. Features such as weather data, time

of day, and day of the week are important predictors of bike demand. In addition, some studies have highlighted the benefits of using ensemble models or hybrid approaches that combine different types of models. From the survey, we understand that a few existing studies focus on batch prediction, neglecting the potential of real-time demand forecasting and dynamic optimization. Addressing the long-term trends and sustainability aspects of bike-sharing demand is underexplored. Developing methodologies to provide probabilistic forecasts could enhance the decision-making process for BSS operators and urban planners. Further, bike demand exhibits complex temporal patterns and dynamics influenced by factors like weather, time of day, day of the week, and seasonality. Investigating traditional and contemporary time-series forecasting models capable of effectively capturing these patterns could present a promising avenue for research. Through this research attempt, we propose to develop forecasting methods that address long-term prediction. Also, we focus on selecting appropriate features that can give better predictions. In contrast with other works, we use a set of fine-grained temporal features for prediction. Overall, the literature suggests that the performance may depend on the choice of model, and additional features, such as weather and working day features, contribute more to the performance of the model, and there is a scope for improvement in prediction.

## 3. Materials and Methods

### 3.1. Dataset Description

In this work, we use datasets that consist of the usage log from the Capital Bike Sharing (CBS) system in Washington, DC, USA [24]. These datasets cover a two-year timeframe and are considered suitable for our research goals due to the following reasons. Firstly, they include at least two full life cycles of the BSS, allowing for the application of supervised and semi-supervised learning methods. Secondly, there are additional sources of data that provide historical information on environmental factors, such as weather conditions, weekdays, and holidays, which can be extracted and incorporated into our analysis. The datasets contain hourly and daily counts of rental bikes, along with the corresponding weather and seasonal information. In this work, we use an hourly dataset that has 17379 instances and 16 attributes such as date, season, year, month, and weather information. Subsequently, we utilized data from the initial 22 months for training purposes, reserving the final two months for testing. This helps to validate the long-term prediction of the proposed models.

### 3.2. Exploring the Data and Outlier Analysis

Exploratory Data analysis involves utilizing statistical and visualization tools to depict data to identify crucial aspects of the dataset for further examination. We examine the dataset from multiple perspectives and present a few visualizations that offer a concise overview. Since the objective of the proposed work is to predict the number of bikes that will be rented shortly, the best way to begin is with the target variable to predict: "count". So, in this work, we stratify the "count" against different predicator variables, such as seasons, months, weather conditions, etc., and show them in Figure 1.

Further, we have also performed outlier analysis to identify the presence of outliers within a dataset. An outlier is a data point that significantly deviates from the normal pattern or distribution of the data. These anomalies can have a significant impact on statistical analysis and modeling results, leading to biased or misleading conclusions. Outliers can disrupt the underlying patterns and trends in bike demand. By detecting and understanding outliers, we can better assess the overall trend and identify whether they are indicative of significant shifts in demand or just noise. This ensures that our predictions are based on a more accurate representation of the underlying demand patterns. Hence, we performed an outlier analysis on a few input variables and presented it in Figure 2.

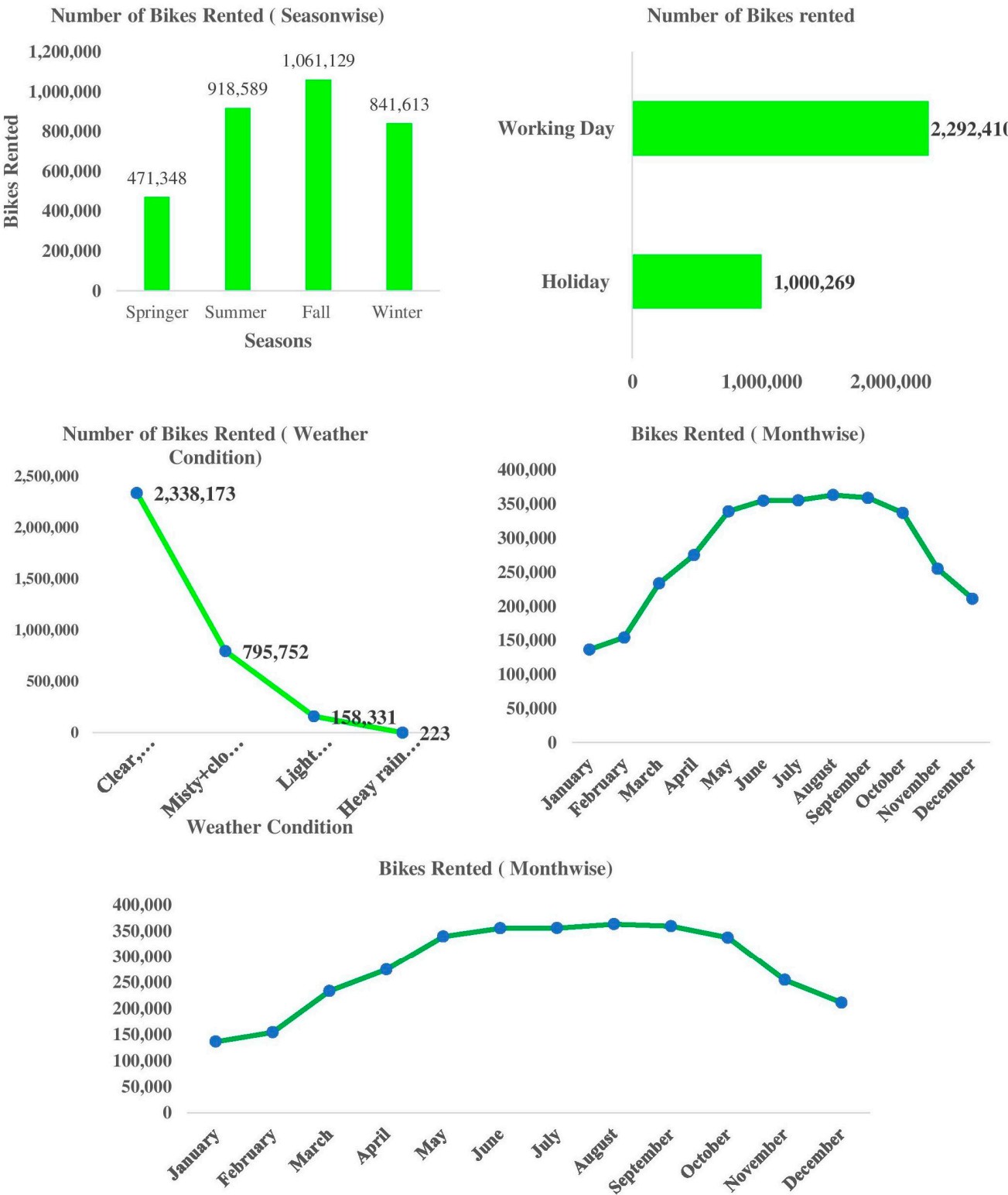

**Figure 1.** Exploratory Analysis of the number of bikes rented on different attributes.

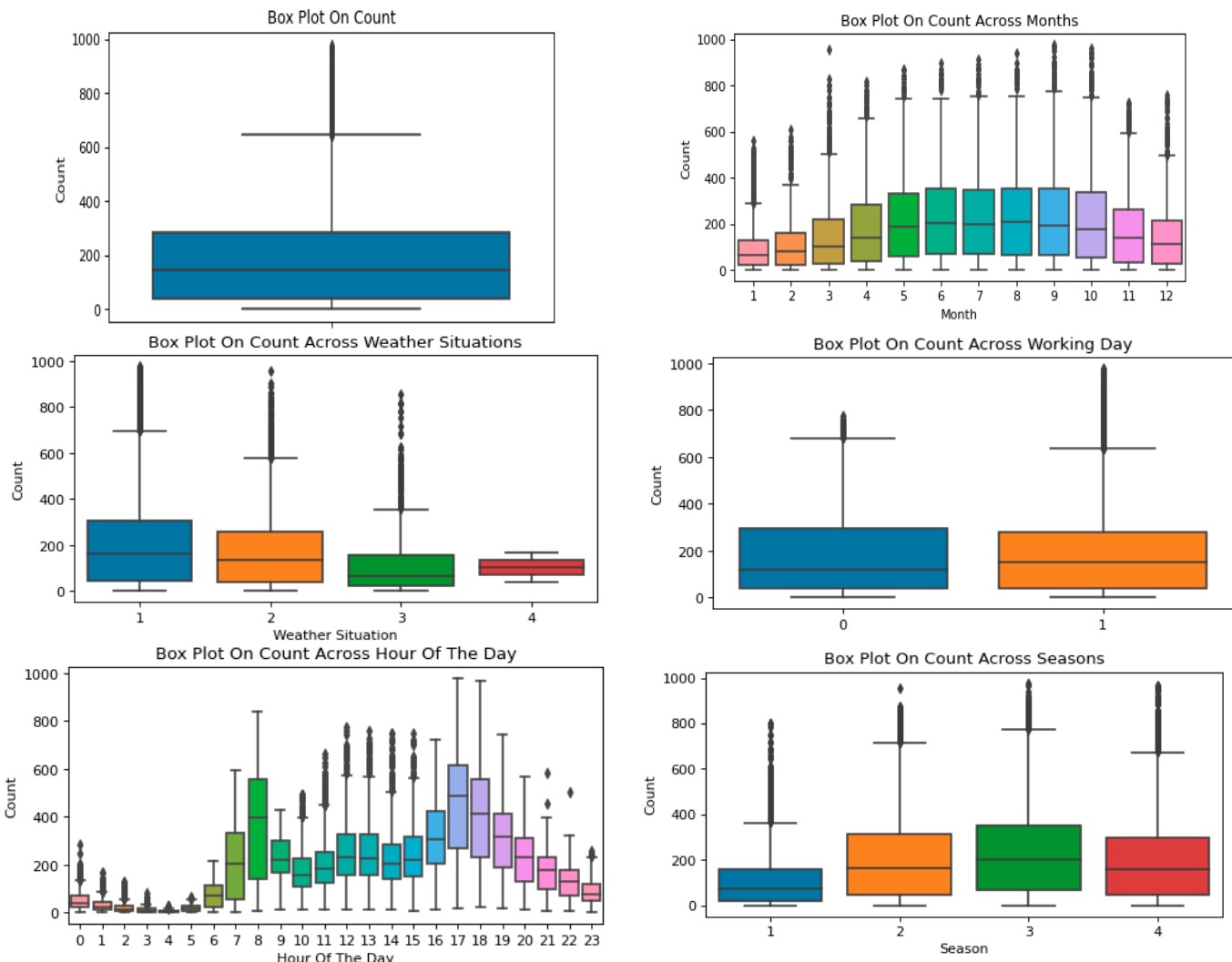

**Figure 2.** Outlier Analysis for the target variable "count".

As seen in Figure 2, the number of bikes rented is between 0 to 1000. For instance, when the weather is extreme, the number of rented bikes is less, and otherwise, its median count increases. Also, the median value increases in the seasons such as summer and fall. Since outliers often represent exceptional or anomalous events that may not reflect the typical demand behavior, we can gain a clearer understanding of the normal patterns and trends in bike demand by removing them. To remove the outliers, we used a common approach called interquartile range (IQR). After removing outliers, we found the correlation between the target and predictor variables. The correlation between different predicator variables and the number of bikes rented is presented in Figure 3. Correlation analysis provides insights into how the predictor variables are related to bike demand. Positive correlations indicate that as the predictor variable increases, the bike demand tends to increase as well. Negative correlations indicate an inverse relationship. Understanding these relationships can help in gaining a deeper understanding of the factors influencing bike demand and in formulating strategies to meet the demand effectively. In addition, we also performed correlation-based feature selection to find the predictors that contribute to bile prediction. From these analyses, we understand that the features like "atemp", "hum", "weathersit", etc., play a less significant role in determining the number of bikes rented. The attributes which play a significant role are explored in Section 4.2.

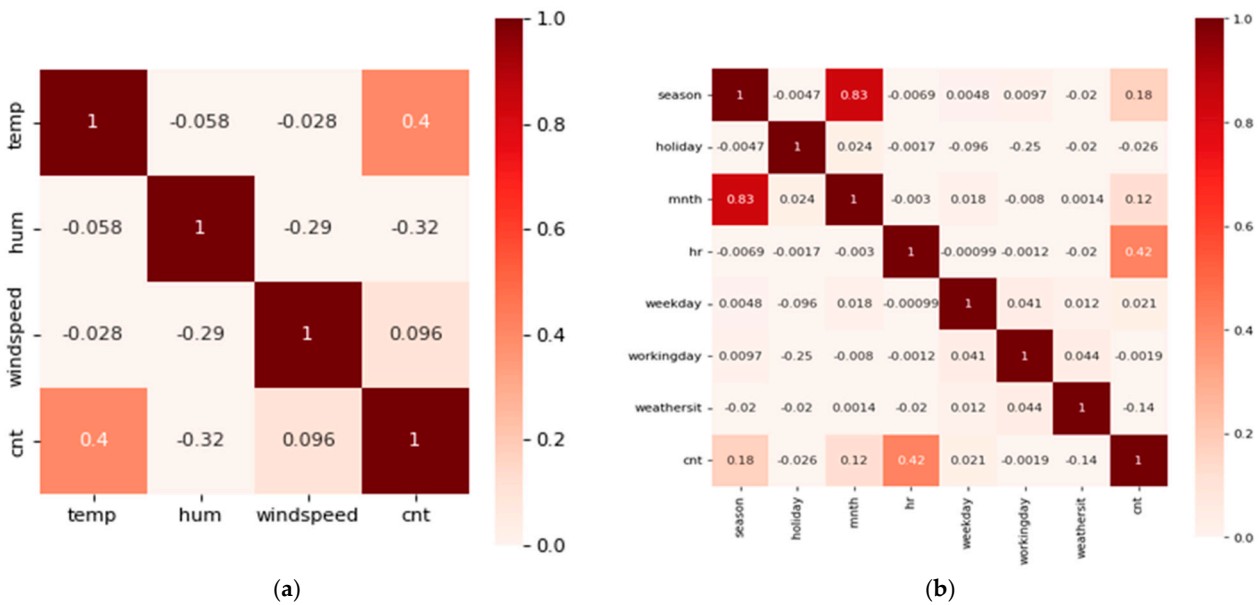

**Figure 3.** (**a**) Correlation between the numerical variables (temperature, humidity, and windspeed) and the count (**b**) correlation between categorical variables (season, holiday, weather, etc.) and the count.

### 3.3. Modeling Approach for Demand Forecasting

In this section, we present a summary of the models developed for predicting the demand for bikes. We use regression and time series models to develop a demand prediction model.

### 3.3.1. Random Forest

Random Forest (RF) creates multiple decision trees on randomly selected subsets of data and features, then combines their predictions to make more accurate predictions [25]. To use this algorithm, we first analyze the importance of each feature and select relevant features that contribute significantly to the bike demand prediction. Then, we train the RF model using the training dataset and the model creates an ensemble of decision trees based on random subsets of features and data samples. During training, we optimize the model's hyperparameters, such as the number of trees and their depth in the forest, and the number of features considered at each split to improve its performance. After training, the model learns patterns and relationships between the input features and the number of bikes rented. Once the model is trained and optimized, we use it to make predictions on test data.

### 3.3.2. ARIMA

Autoregressive Integrated Moving Average (ARIMA), a time series forecasting model [26], is used to model time series data and make predictions. ARIMA models require the time series to be stationary, meaning that the mean and variance should be relatively constant over time. To address the non-stationarity of the dataset utilized in this study, differencing is employed as a technique to achieve stationarity. Differencing involves taking the difference between consecutive observations to remove trends or seasonality. Figure 4 shows the non-stationary and stationary (by differencing) datasets. Each of the AR, I, and MA components are included in the model as parameters such as p, d, and q. An Autocorrelation Function (ACF) plot is used to determine the appropriate values for p and q. The ACF plot with 50 lags is shown in Figure 5.

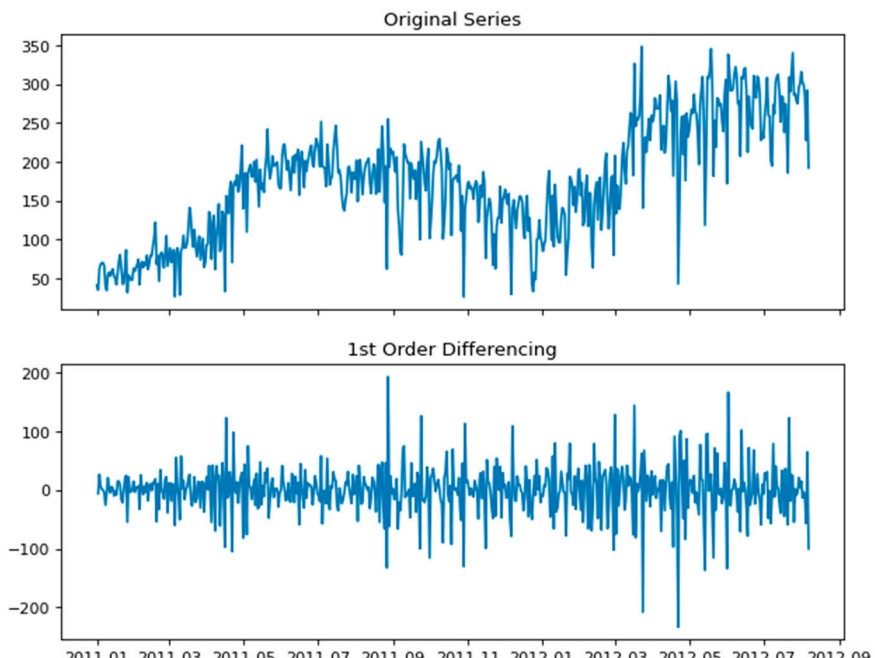

**Figure 4.** First Order Differencing (stationary) of the dataset.

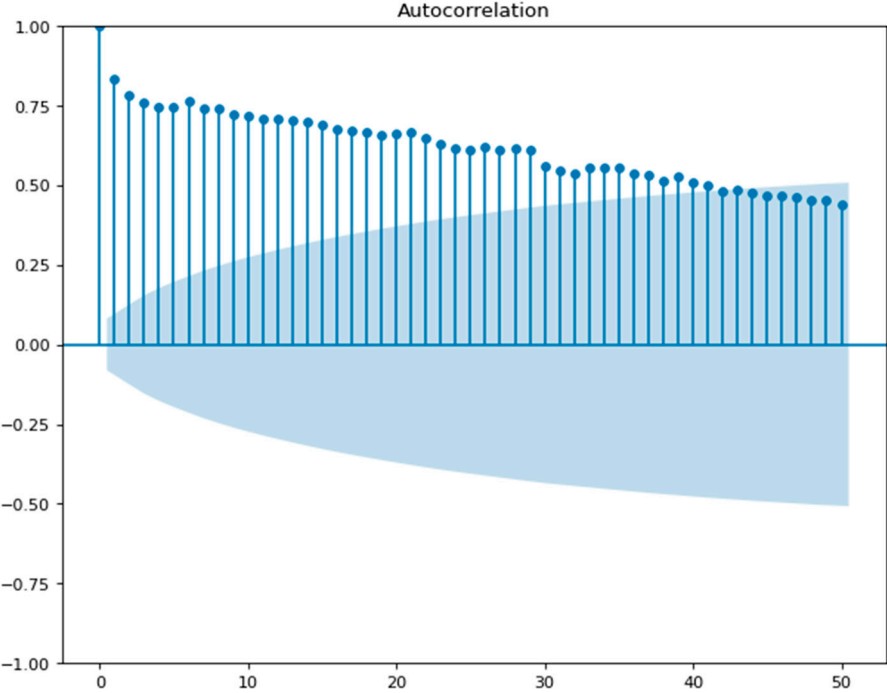

**Figure 5.** Autocorrelation Function plot to find p and q values.

The ACF measures the correlation between the time series and its lagged values. The ACF plot displays the correlation coefficients for different lag values.

- If the ACF plot shows a gradual decline and becomes statistically insignificant after a few lags, it suggests an AR component. The lag at which the ACF plot crosses the significance level for the first time indicates the value of p for the AR component.
- If the ACF plot exhibits a significant spike at a specific lag followed by a sharp drop, it suggests a MA component. The lag at which the spike occurs in the ACF plot indicates the value of q for the MA component.

Using ACF, proper values of p and q are found. Then, the model is trained using a training dataset. If the ARIMA model does not perform satisfactorily, we can refine it by adjusting the model order. ARIMA models are particularly useful when the historical patterns of bike demand exhibit autocorrelation (dependence on past observations) and/or seasonality.

### 3.3.3. SARIMA

Seasonal Autoregressive Integrated Moving Average (SARIMA) is an extension of ARIMA that is used for seasonal time series data [27]. SARIMA models can handle data with seasonal patterns and these parameters help to capture the seasonal fluctuations in the data. Since the dataset exhibits seasonal patterns, we use this model to forecast the demand for bikes.

The following are the steps used to develop the prediction model using SARIMA:

1. Analyze the data for any trends, seasonality, or other patterns.
2. Determine the appropriate values for p, d, q (non-seasonal components), P, D, Q, and S (Seasonal SARIMA components) based on data analysis and ACF plots.
3. Fit the SARIMA model using the training data.
4. Evaluate the model's performance on the test set using appropriate metrics.
5. Fine-tune the model by adjusting the parameter values or trying different combinations.
6. Make predictions for future periods using the trained model.

By utilizing the SARIMA model, we can capture both the non-seasonal and seasonal components of the bike demand data, enabling us to make accurate predictions.

### 3.3.4. LSTM

Long Short-Term Memory (LSTM) is commonly used for time series forecasting [26,27] and is particularly effective at modeling sequences with long-term dependencies. LSTM models are designed to overcome the limitations of traditional neural networks in capturing long-term dependencies in sequential data. Since LSTM can identify recurring patterns in bike demand, such as daily or weekly trends, and use this information to make accurate predictions for future time steps, we intend to use LSTM also. Since the demand patterns can be influenced by multiple factors, such as weather conditions, holidays, seasons, etc., LSTM can capture these complex relationships, allowing it to generate more accurate forecasts compared to traditional linear models. In the proposed bike demand forecasting, LSTM utilizes historical demand sequences to understand how past patterns and trends relate to future demand.

### 3.3.5. GRU

Gated Recurrent Unit (GRU) is similar to LSTM but has fewer parameters [28]. Similar to LSTM, GRU models are designed to capture long-term dependencies in sequential data. This is crucial for bike demand forecasting, as it allows the model to understand and leverage patterns and trends that occur over longer time horizons, such as weekly or monthly cycles. GRU models can adapt to changing patterns in the dataset. They can quickly update their internal state based on new information, allowing them to capture shifts in bike demand patterns due to various factors. This adaptability is beneficial for forecasting bike demand shortly, where patterns may evolve over time.

From the existing research attempts, we find that most of them focus on short-term prediction rather than long-term prediction. As BSS aims for sustainable growth, understanding long-term demand trends is essential to scale the system effectively. Such predictions help in planning the expansion, relocation, or reduction of stations and resources based on projected demand growth or shifts over time. Further, the proposed work aims to find the optimal set of fine-grained features for demand prediction. These features help the models recognize and account for temporal patterns, such as daily and weekend commuting peaks, and seasonal variations. Figure 6 illustrates the comprehensive procedure for forecasting shared bike demand. Beginning with the loading of the bike

dataset, the subsequent steps encompass data pre-processing, followed by the construction of prediction models and a subsequent comparison of their performance.

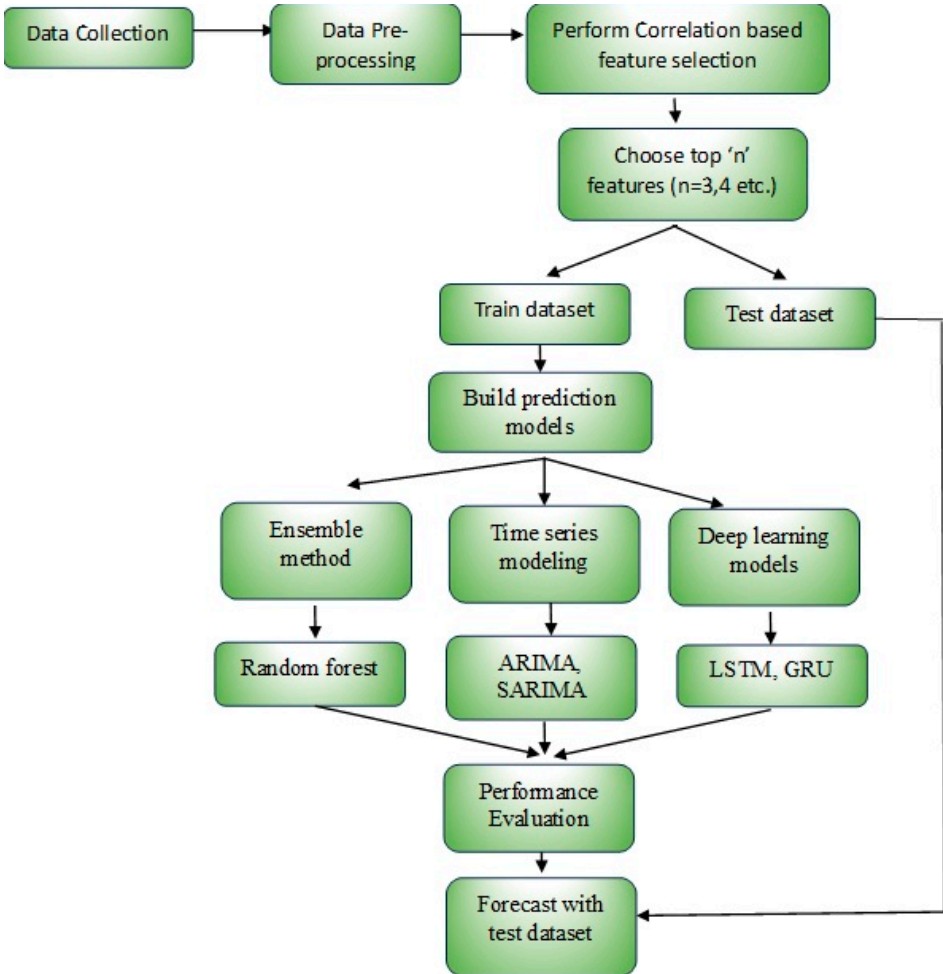

**Figure 6.** Proposed workflow.

## 4. Experiments and Results

### 4.1. Performance Metrics

After training the models, we evaluate the performance of the trained models using the testing dataset. The choice of performance metrics for time series forecasting depends on the specific problem and the goals of the analysis. Common evaluation metrics for regression tasks such as time series prediction include Mean Squared Error (MSE), Root MSE (RMSE), and Mean Absolute Error (MAE).

MSE: MSE is computed by taking the average of the squared differences between the predicted values and the actual values. MSE penalizes larger errors more heavily due to the squaring operation. MSE is less sensitive to outliers than RMSE, making it a better choice when outliers are present in the data.

MAE: MAE is determined by calculating the average of the absolute squared differences between the predicted and the actual values. MAE does not penalize larger errors as heavily as MSE.

RMSE: RMSE on the other hand, is calculated by taking the square root of the mean of the squared differences between the anticipated and observed values. RMSE is more sensitive to outliers than MAE, making it a better choice when outliers are not present in the data. The smaller the MAE, MSE, and RMSE value is, the higher the prediction accuracy and the stronger the feature expression ability of the model.

### 4.2. Experimental Settings and Results

#### 4.2.1. Experimental Settings

The proposed work has been implemented using Python under Google Colab. For developing the models, packages like statsmodel, sklearn, and keras have been imported and appropriate functions have been used. The selection of predictor variables that have an impact on the target variable (the number of bikes rented) was found using the correlation-based feature selection. The correlation shows which variables, such as temperature, hour, holiday, humidity, season, wind speed, weekday, and weather, best fit the data and forecast appreciably. All the prediction models have been trained using the training dataset and their performance has been tested against the testing dataset. The results of evaluating the performance of each model using the metrics discussed in Section 4.1 are presented in this section.

The dataset is split for the experiment in an 80:20 ratio as training and testing datasets, respectively. First, we ran the RF model over the training dataset with all the input features and then, the test dataset has been evaluated using the model. The performance of the model based on RF has been calculated using different hyperparameter settings such as n_estimators, max_features, max_depth, min_samples_split, etc. Table 1 shows the settings of hyperparameters for the RF model. We use randomized search to find the optimal values of hyperparameters which give better results. The optimal values for the hyperparameters are also presented in Table 1.

**Table 1.** Search Space for the hyperparameters in the RF model.

| Hyperparameters | Search Space | Optimal Value |
|---|---|---|
| n_estimators | [400, 500, 700, 800, 1000, 1300, 1600, 1900, 2000] | 1600 |
| Max features | ['auto', 'sqrt'] | auto |
| Max depth | [None, 10 to 110 in steps of 10] | 90 |
| Min samples split | [2, 4, 5, 8, 10 ] | 5 |
| Min samples leaf | [1, 2, 4, 8] | 1 |
| Bootstrap | [True, False] | True |

For the ARIMA model, the values of p, q, and d play a vital role and are determined using the auto-correlation function as explained in Section 3.3.2. Similarly, the appropriate values for p, d, q (non-seasonal components), P, D, Q, and S (Seasonal components) for the SARIMA model are determined by ACF plots. For LSTM and GRU-based models, the number of layers, number of neurons per layer, and activation functions are fine-tuned.

#### 4.2.2. Experimental Results

The dataset has 13 input features, and all these attributes may not significantly contribute to finding the number of bikes being rented. So, we perform a rigorous analysis of these attributes to determine their role in prediction. First, we used all the input features for training and testing. The prediction performance of all the developed models for the test dataset is presented in Table 2.

**Table 2.** Results of the models for all input features.

| Metrics | RF | ARIMA | SARIMA | LSTM | GRU |
|---|---|---|---|---|---|
| MSE | 5155.89 | 7258.02 | 5802.4 | 3242.16 | 3188.86 |
| RMSE | 71.80 | 85.19 | 76.17 | 56.94 | 56.47 |
| MAE | 44.49 | 64.09 | 56.35 | 35.21 | 33.76 |

Figure 7 depicts the actual and predicted number of bikes rented for the test dataset using the developed models.

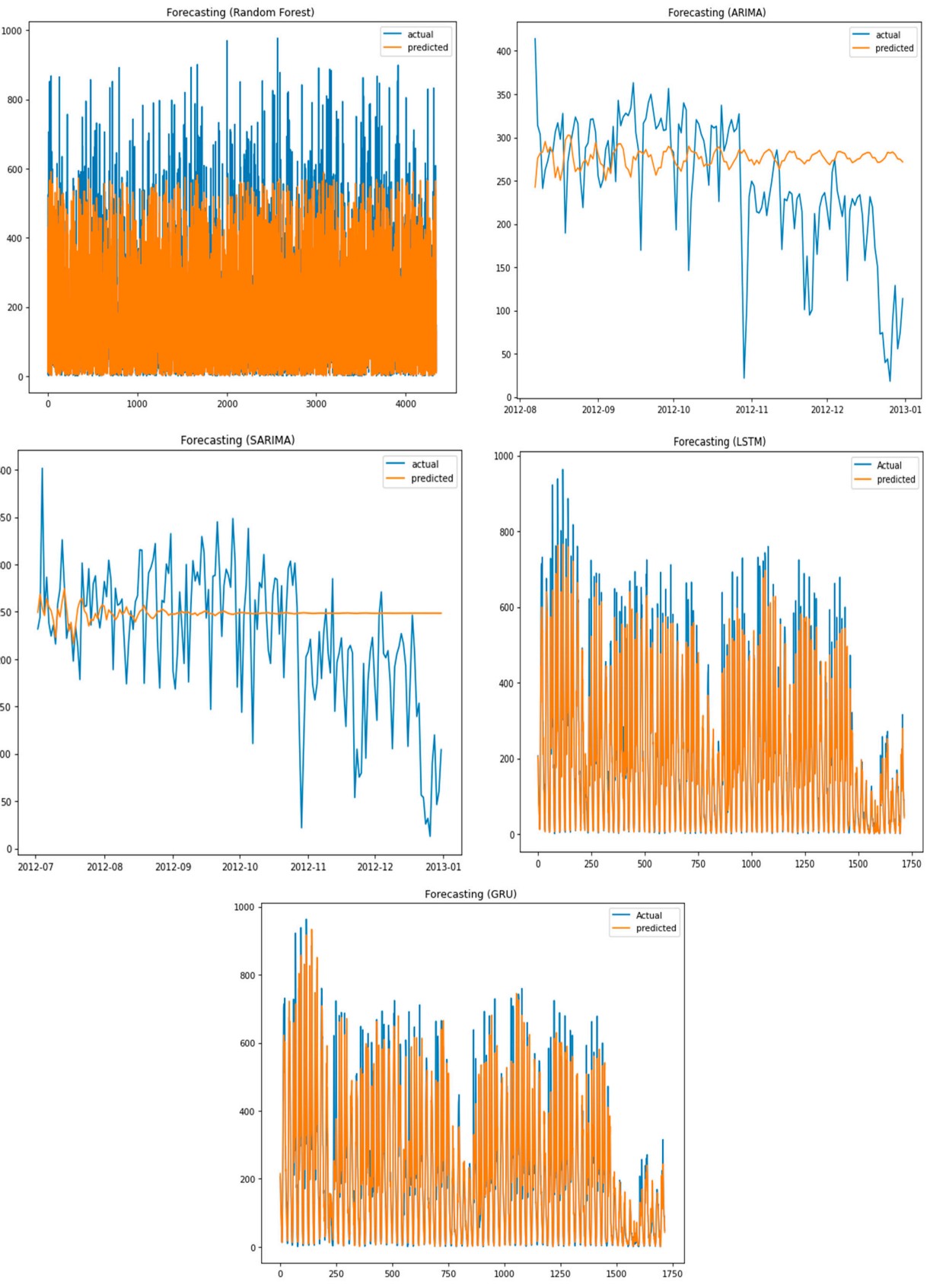

**Figure 7.** Actual vs. forecast for test data.

We initially trained all the models using the training datasets with all input features and the results are presented above. But, from the correlation function, we understand that all the input features do not play a significant role in forecasting the bike demands. Hence, we identified and removed a few sets of attributes using various techniques such as p-values and correlation-based feature selection methods. And, then we trained the models and evaluated the models without insignificant features. The results of such training are presented in Table 3, which also shows the set of attributes used for forecasting.

**Table 3.** Results of the models with different sets of features selected for testing.

| Attributes Selected | Models | MSE | RMSE | MAE |
|---|---|---|---|---|
| month, hour, weekday (Feature Set 1) | RF | 9893.978 | 99.47 | 63.948 |
| | ARIMA | 4371.854 | 66.12 | 54.712 |
| | SARIMA | 5801.870 | 76.17 | 56.354 |
| | LSTM | 3316.49 | 57.589 | 35.381 |
| | GRU | 3120.22 | 55.859 | 33.281 |
| month, hour, weekday, year (Feature Set 2) | RF | 6120.259 | 78.232 | 44.392 |
| | ARIMA | 7258.02 | 63.62 | 52.493 |
| | SARIMA | 5802.37 | 68.52 | 54.653 |
| | LSTM | 3582.50 | 59.854 | 36.084 |
| | GRU | 2676.82 | 51.738 | 31.875 |
| month, hour, weekday, year, season (Feature Set 3) | RF | 5625.067 | 75.00 | 42.367 |
| | ARIMA | 7258.02 | 71.19 | 50.521 |
| | SARIMA | 5802.37 | 73.84 | 56.356 |
| | LSTM | 3979.34 | 63.082 | 38.207 |
| | GRU | 3646.95 | 60.39 | 35.584 |
| month, hour, weekday, year, season, holiday, working day (Feature Set 4) | RF | 5062.746 | 71.15 | 39.926 |
| | ARIMA | 4162.959 | 64.521 | 37.231 |
| | SARIMA | 3966.984 | 62.984 | 35.956 |
| | LSTM | 3713.07 | 60.935 | 36.973 |
| | GRU | 2641.24 | 51.393 | 30.764 |
| month, hour, weekday, year, season, holiday, working day, weathersit and temp (Feature Set 5) | RF | 3123.477 | 55.89 | 30.360 |
| | ARIMA | 3508.311 | 59.231 | 36.621 |
| | SARIMA | 3493.164 | 59.103 | 36.001 |
| | LSTM | 3381.42 | 58.15 | 35.890 |
| | GRU | 3276.53 | 57.241 | 34.500 |

## 5. Findings and Discussion

The proposed research work explored the temporal dependencies of travel demands using regression and time series models. To identify the correlation between the predictor and target features, correlation analysis has been performed. The study focused on two traditional time-series forecasting models (ARIMA, SARIMA) deep learning methods (LSTM GRU), and an ensemble method (Random Forest). In this section, we discuss the performance of the developed models, and interpret and present our findings. We trained and ran the models using different sets of input features. Initially, we ran using all the features without considering their correlations to the bike demand and found that GRU has performed well by giving lower MSE and MAE values. This is shown in Figure 8.

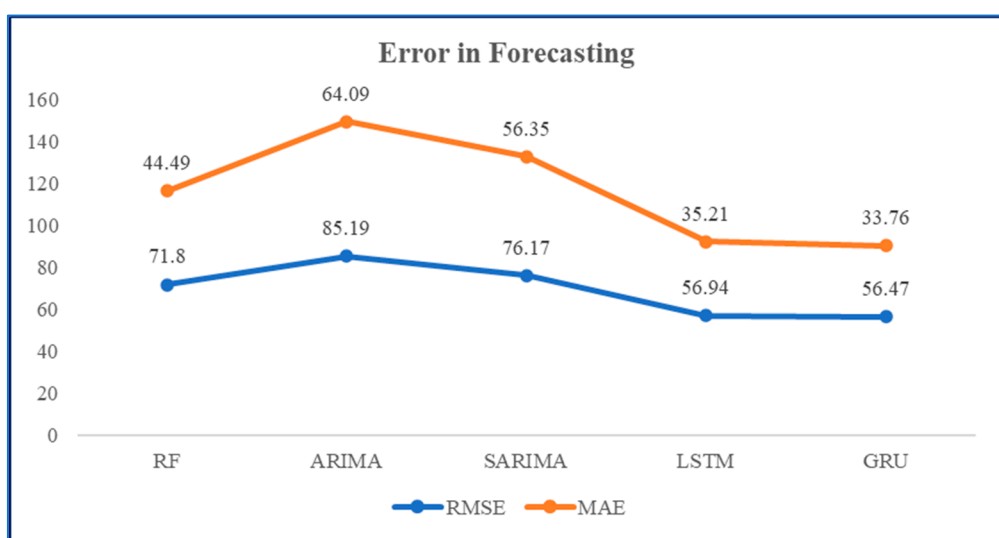

**Figure 8.** Performance in terms of RMSE and MAE (with all input features).

Then, we ran the models with selected features, and these features have been selected using a correlation-based selection method. First, we took "mnth", "hr" and "weekday" attributes as they stand at the top of the list identified by the feature selection method. Then, gradually we added attributes to train the models. The results are shown in Table 3. We also present the same in Figures 9 and 10 for better visualization. From these figures, we can see that GRU performs comparatively better for feature sets 1, 2, 3, and 4. However, RF performs better for feature set 5 in terms of MAE. For feature set 4, GRU gives a low MSE/RMSE value. While analyzing the reasons for such performance, we find that RF performs better because it is good at handling both numerical ("temp") and categorical features ("hr", "mnth", "season", etc.) and can capture complex relationships in the data. Hence, we understand that RF can be effective for bike demand forecasting, especially when there are multiple relevant categorical and numerical features available. However,, GRU also performs equally better for most of the feature sets. Since the gating mechanisms of GRU enable it to capture long-term dependencies in time series data, which is important for bike demand forecasting, GRU leads to better prediction. As biking patterns and demands are influenced by various factors that might have delayed effects, GRU has effectively modeled these dependencies.

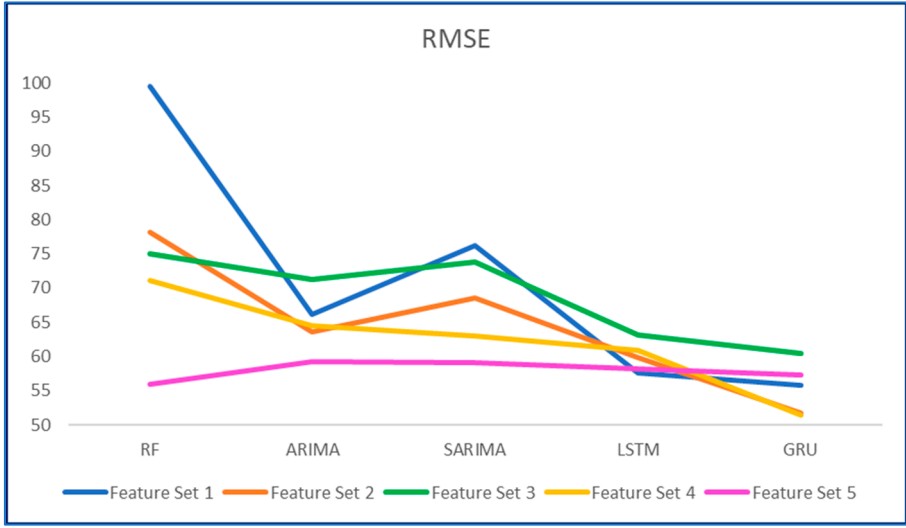

**Figure 9.** RMSE for the different sets of features.

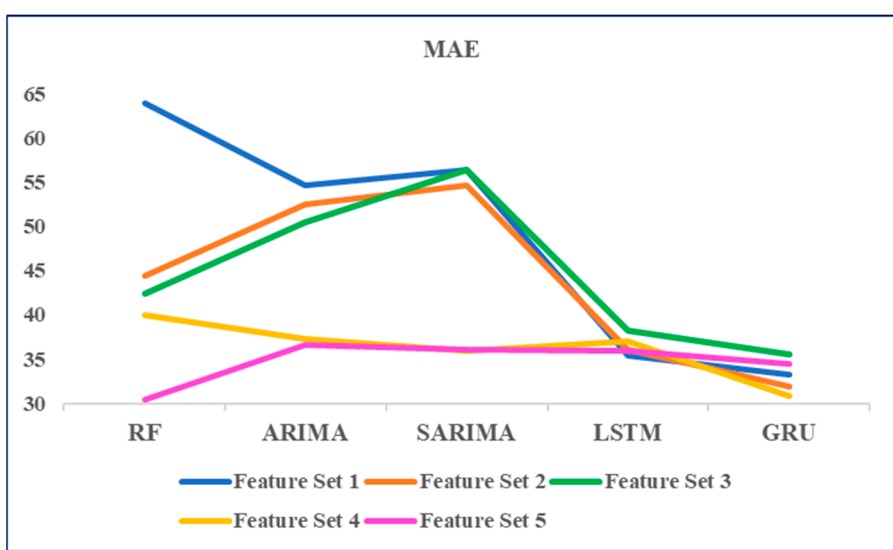

**Figure 10.** MAE for the different sets of features.

GRU can also work effectively with numerical features, especially if those features are relevant and informative for predicting bike demand. Numerical features might include historical bike demand data, temperature, humidity, or other factors that influence demand patterns. In many real-world scenarios, a combination of both categorical and numerical features is used to capture the complexity of bike demand patterns accurately. The GRU model, being capable of handling both sequential data and additional features, can leverage this information to improve its forecasting performance. Compared to deep learning and ensemble models, there is a significant deviation between actual and predicted variables. Since the ARIMA and SARIMA models are designed to capture linear dependencies in the time series data and the relationship between the target variable and the predictors is non-linear, these models may not be able to capture the underlying patterns effectively, leading to higher prediction MAE and MSE.

In this research, we employed historical bicycle data to conduct long-term forecasting, and this was achieved by leveraging various sets of features outlined in Table 3. For long-term prediction, we have taken the instances of the last two months of the datasets as the testing data and predicted the demand. We have evaluated the performance of the different time series modeling algorithms and compared them to actual demand and presented in Table 3. In addition, we have fine-tuned the process of prediction by choosing a different set of features. These features allow the model to capture variations in demand that occur at different times of the day and different days of the week. This enables the model to identify recurring patterns and trends, such as daily commuting patterns, working day and week-end patterns, and off-peak periods. The inclusion of time-related features adds granularity to the understanding of demand fluctuations. By identifying relevant features, the risk of overfitting, the model capturing noise rather than actual patterns, is reduced. The selected features contribute to a more robust and generalizable model.

Still, there are a few unsettled points in dynamic bike demand prediction that require further exploration. A few potential limitations in dynamic bike demand prediction include:

i. Special events and occasions can influence bike demand.
ii. Changes in infrastructure, such as new bike lanes or changes in public transportation routes, can influence bike demand patterns.
iii. Latency in real-time or near-real-time predictions for dynamic bike demand
iv. Unforeseen events, such as road closure for maintenance, natural calamities, public health crises (like COVID-19), etc., and social-environmental issues, such as equity and accessibility, can disrupt regular demand patterns.

These points might involve challenges, uncertainties, or areas where more investigation is needed. In addition, there are some methodological challenges to the proposed models which are listed below:

i.   Data from a single BSS might not be representative of the bike demand patterns of other locations in the city. It could be biased toward specific user demographics, usage patterns, or geographic locations.

ii.  A BSS in one location may not exhibit the full range of demand patterns that occur across different lotions of a city.

iii. Bike demand patterns may change over time due to various factors, including changing user behaviors, weather patterns, and urban developments. The models trained on historical data might struggle to adapt to these evolving patterns.

To check whether the developed models can address the above challenges, we have taken another dataset named the London Bike Sharing Dataset [29] which contains the bike sharing details collected between January 2015 and December 2016. Various temporal features, such as temperature, season, weather details, wind speed, etc., and predicted bike demand have been recorded in the dataset and contains 17,000 observations approximately. The results of the predictions of the proposed models against this dataset are presented in Table 4.

**Table 4.** Performance of the models on the London Bike Sharing Dataset.

| Metrics | RF | ARIMA | SARIMA | LSTM | GRU |
|---------|---------|---------|---------|---------|---------|
| MSE | 5992.64 | 8413.57 | 6319.83 | 4302.41 | 3965.34 |
| RMSE | 77.41 | 91.72 | 79.49 | 65.59 | 62.97 |
| MAE | 52.03 | 70.92 | 61.27 | 39.52 | 35.22 |

Without fine-tuning, we executed the developed models against the London Bike Sharing Dataset. The results indicated that, while the performance was not exceptional, it was not poor. However, we believe that, through the utilization of diverse datasets and suitable feature selection techniques, transfer learning methodologies, and the integration of predictions derived from models trained on various BSS datasets, performance can be enhanced, and a broader spectrum of demand patterns can be captured. While generalizing the models, there will always be some level of fine-tuning and continuous evaluation of models required for each BSS.

## 6. Conclusions

In this study, we explored and compared five different forecasting models, namely RF, ARIMA, SARIMA, LSTM, and GRU, for predicting bike demand. Before forecasting, we performed an exploratory data analysis to find the correlation between the predictor and target features. Based on the correlation results, we trained and tested the models using different sets of features and found that GRU performs better for different sets of features with the least RMSE value of 51.393. Surprisingly, RF performed better when considering both numerical and categorical features with the least MAE value of 30.36. Through this study, it is demonstrated that the temporal features are crucial for forecasting the demand for bike-sharing. In conclusion, the selection of the best forecasting model for bike demand prediction depends on the specific dataset and the forecasting horizon.

While our research offers insightful information on the effectiveness of various forecasting algorithms, for bike demand prediction, there are several avenues for future work to improve the accuracy and robustness of the predictions. Although this study considers temporal dependencies, several spatial factors shall also be considered in future work. Careful selection and engineering of relevant features can enhance the models' ability to capture underlying patterns. Further, we intend to develop methods to incorporate real-time data streams to make more up-to-date and accurate predictions for bike demand.

**Author Contributions:** Conceptualization, M.S. and A.M.; methodology, S.V.E.; software, A.M.; validation, R.C. and M.S.; formal analysis, R.C. and A.M.; resources, J.C.; data curation, R.C.; writing—original draft preparation, M.S.; writing—review and editing, A.M.; visualization, R.C.; supervision, J.C.; project administration, S.V.E.; funding acquisition, J.C. All authors have read and agreed to the published version of the manuscript.

**Funding:** This work was supported by the Institute of Information and Communications Technology Planning and Evaluation (IITP) funded by the Korean Government, Ministry of Science and ICT (MSIT) (Implementation of Verification Platform for ICT Based Environmental Monitoring Sensor), under Grant 2019-0-00135 and partially supported by the Institute of Information and Communications Technology Planning and Evaluation (IITP) funded by the Korea Government, Ministry of Science and ICT(MSIT) (Building a Digital Open Lab as open innovation platform) under Grant 2021-0-00546. This research was supported by "Research Base Construction Fund Support Program" funded by Jeonbuk National University in 2023.

**Institutional Review Board Statement:** Not applicable.

**Informed Consent Statement:** Not applicable.

**Data Availability Statement:** Not applicable.

**Acknowledgments:** We would like to acknowledge our sincere thanks to K.S. Danushraj and V. Gowtham for having extended their cooperation in developing the code.

**Conflicts of Interest:** The authors declare no conflicts.

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
