# Peer review of "Enhancing Sustainable Transportation: AI-Driven Bike Demand Forecasting in Smart Cities"

_sustainability, doi:10.3390/su151813840_

Round 1

Reviewer 1 Report

This paper examines various time series and machine learning algorithms to forecast bike demand in Sustainable Bike-Sharing Systems. It compares LSTM, GRU, RF, ARIMA, and SARIMA models using historical data from the Capital Bikeshare system in Washington D.C. The findings reveal that GRU yields the most accurate predictions of bike demand, followed by RF. ARIMA and SARIMA models produce less precise results, likely due to their assumptions. These insights are crucial for future investigations and decision-making in bike-sharing systems. Overall, this paper is well-structured and of great value. Nevertheless, there are a few issues that should be acknowledged and addressed.

1.In lines 80-85, the authors discuss the scope of the problem under investigation. However, the problem statement appears to be too broad and is not fully addressed in this paper. To improve the clarity and focus of the research, it is essential for the authors to provide a more specific and targeted explanation of the problem, considering that Bike Demand Forecasting is a widely studied topic.

2. In the Literature Review section, it is highly recommended to highlight the gaps or unanswered questions that exist in the current body of research. This helps to establish the significance and relevance of your study. Discuss how your research aims to fill these gaps or address these unresolved questions, demonstrating the unique contribution your study will make to the field.

3. In the Materials and Methods section, it is important to clearly state the main contribution of the proposed method. Describe how your approach differs from existing methods or how it improves upon them.

4. In the Materials and Methods section, it is not necessary to provide detailed descriptions of well-known techniques or methods such as LSTM or GRU. Instead, it is sufficient to reference the relevant literature where these methods have been adequately described. This allows you to focus on describing the aspects of your own methodology that are unique or different from existing approaches, thereby highlighting the main contributions of your study.

5. To improve the figure captions, it is recommended to provide more specific and informative details in each caption.

6. Despite the numerous proposed experiments, it is crucial to thoroughly discuss and interpret the experimental results. Analyze trends, patterns, exceptions, and unsettled points in detail. Ensure a clear alignment with research objectives, thereby enhancing the overall understanding and credibility of the study.

Author Response

Please find the report attached

Reviewer 2 Report

Review report of Sustainability-2566395: Enhancing Sustainable Transportation: AI-Driven Bike Demand Forecasting in Smart Cities

Recommendation    Major revision

General comments

This paper is motivated by the need for accurate bike demand forecasting to optimize bike-sharing systems and promote sustainable transportation in smart cities. It aims to compare the performance of different time series and machine learning algorithms for predicting bike demand using a two-year historical log from the Capital Bikeshare system in Washington, D.C. The paper reviewed existing works related to bike-sharing systems (BSS) and demand forecasting. The dataset used is then described, which includes hourly bike rental records from 2011 to 2012 and weather and calendar data. An exploratory data analysis is presented, which includes visualizations of bike demand patterns and correlations with weather and calendar variables.

The performance of different time series and machine learning algorithms for predicting bike demand are compared. The time series models include ARIMA and SARIMA, while the machine learning models include random forest (RF), LSTM, and GRU. The models are evaluated using various metrics, including mean absolute error (MAE), root mean squared error (RMSE), and mean absolute percentage error (MAPE). The results show that the machine learning models generally outperform the time series models regarding prediction accuracy, and GRU achieves the best performance among the machine learning models. The authors also perform feature importance analysis to identify the most important variables for predicting bike demand, which includes temperature, humidity, wind speed, and hour of the day. Finally, this study concluded that accurate bike demand forecasting can help optimize bike-sharing systems by improving bike availability and reducing operational costs. It also highlighted the importance of incorporating weather and calendar data in bike demand forecasting models, as these variables significantly impact bike demand patterns.

Overall, this work demonstrated the effectiveness of machine learning models for predicting bike demand and highlight the importance of considering weather and calendar variables in bike demand forecasting models. This paper tends to outline the importance of machine-learning techniques in forecasting but lacks many details, which require major revision.

Detailed comments

1.      The authors should recheck the journal’s requirement for the citation style, as full-name citation is always odd in international journals. A correct form example in the paper should be Box et al. [15]; a wrong form example in the paper is Rob J. Hyndman et al. [16]. Furthermore, ‘et. al.’ should be ‘et al.’.

2.      The contributions should be carefully rewritten. The reviewer agrees with the research questions outlined by the authors but not the contributions. The reviewer does not agree that the exploratory analysis and literature review can be recognized as contributions, given that this study is forecasting-related. Moreover, the remaining two contributions are also unclearly described: ‘analyze the impact of temporal factor’ is too broad, and ‘develop forecasting model’ is not what the work performed, given that the details of all forecasting methods are not clearly tractable.

3.      As a manuscript accompanied by numerous methods, Section 3.3 loses many details, so readers are hard to follow; for example, RF and LSTM.

4.      The submission should clearly introduce the programming language or libraries adopted.

5.      The authors should explain why only MAE, RMSE, and MAPE are used as the measuring metric, as outliers or skewed data may influence them.

6.      For the research applicability, the authors are recommended to include several more limitations. These include other potential spatiotemporal factors influencing bike demand (e.g., special events, road closures, or public transportation disruptions) and social-environmental issues (e.g., equity, accessibility, or safety).

The English writing style is satisfactory.

Author Response

Please find the report attached

Reviewer 3 Report

Thank you for the opportunity to review this paper on bike sharing balancing. Overall, the paper is well written and an interesting to read. A few minor questions and comments for the authors:

What time scale does the CaBi data set used for this study cover?

What are the methodological limitations, including any limitations associated with leveraging data from a single operator/location. Do the authors know if the findings would have been similar if another location or operator had been used? As such, can the authors comment on the potential generalizability of the paper's results. 

No comments.

Author Response

Please find the attached report

Reviewer 4 Report

The research paper does provide some scientific literature analysis. However, the research gap needs to be more clearly formulated. Having said that, I encourage the authors to formulate the gap addressed by the current research and explain in detail how the research questions are appropriate to close this gap. 

The authors present the empirical results by using different approaches. I encourage the authors to use the same way of presentation and to add all necessary explanations (title of the axis, the title of columns, etc.).

The methodology chosen could be considered appropriate, and no further improvements are necessary. The research sample is limited, so one could question the results' significance. Moreover, it is not clear why the particular sample was chosen. I encourage the authors to address these concerns and to provide tests that the results gained and conclusions are not biased. 

Please review the text once again for clarity purposes, to make it more reader-friendly.

Author Response

Please find the attached report

Round 2

Reviewer 2 Report

General comments

The reviewer agrees with most replies on the revised manuscript. Nevertheless, some minor issues are still required to be amended by the authors.

New comments

1.     Some tables and figures have missed titles or are empty. This hinders the understanding of the manuscript (e.g., Table 1 on Page 13; Figure 6 on page 12).

2.     Some paragraphs have wrong formats (e.g., Lines 383-387; 495-497).

3.     The authors should offer brief descriptions of the London Bike Data (at least about the duration (starting and ending dates) and the numbers), given that the proposed model in this submission requires considerable data. 

The English is understandable.

Author Response

We would like to extend our heartfelt thanks to the reviewer for his meticulous evaluation of our manuscript and the feedback and suggestions provided. We have incorporated the comments into our manuscript in response to the recommendations.

Reviewer 4 Report

Thank you for considering the improvement suggestions. Good luck with your research!

Please check the formatting of the article once again. 

n/a

Author Response

We would like to express our sincere gratitude to the reviewer for the through analysis of our paper as well as the comments and recommendations.  We have incorporated all the suggestions given by the reviewer. 
